# Beyond antibodies: Beta-2 glycoprotein I as the unsung guardian of pregnancy

**Kazunobu Yagi, Reina Komatsu, Hitomi Nakamura** *, **Kazuya Mimura**,
**Masayuki Endo, Takuji Tomimatsu, Tadashi Kimura**

Department of Obstetrics and Gynecology, Osaka University Graduate School of Medicine, Suita, Osaka, Japan

* hitomi@gyne.med.osaka-u.ac.jp

## Abstract

The physiological function of beta 2 glycoprotein I (β2GPI) itself is not well understood, other than that it is a primary antigen to anti-phospholipid antibodies in the autoimmune disease antiphospholipid syndrome. β2GPI is a soluble blood protein that is predominantly synthesized in hepatocytes. Why is the expression of β2GPI observed in the placenta despite its abundance in the circulating blood of healthy individuals? Does the placenta produce a specific-acting β2GPI?. β2GPI was recently shown to adopt two interconvertible biochemical confirmations based on the integrity of disulfide bonds: oxidized and reduced. The present study investigates the physiological function of β2GPI in trophoblast cells, with a focus on the reduced and oxidized forms of β2GPI under the hypothesis that placental β2GPI has a different activity from circulating β2GPI. Endogenous β2GPI secretion in trophoblast cells were predominantly in the reduced form, while those in HepG2 liver cells were mainly in the oxidized form. Progesterone increased reduced-β2GPI in both the trophoblast and liver cells. Oxidized-β2GPI significantly inhibited trophoblast cell migration and increased placental soluble fms-like tyrosine kinase-1 (sFlt-1). Furthermore, excess sFlt-1 significantly increased oxidized-β2GPI secretion in HepG2 cells. Circulating oxidized-β2GPI levels were significantly higher in women with pre-eclampsia than in those without pre-eclampsia. Therefore, oxidized-β2GPI may contribute to the pathogenesis of pre-eclampsia. Under oxidative stress, the excessive oxidation of β2GPI and/or excessive placental sFlt-1 may trigger a negative spiral between trophoblast and liver cells.

## Introduction

The anti-beta 2 glycoprotein I (β2GPI) antibody is well-known as the main anti-phospholipid antibodies (aPL) that characterize the autoimmune disease anti-phospholipid syndrome (APS). Besides its role as a primary antigen to aPL in APS, the physiological function of β2GPI (also known as apolipoprotein H) remains poorly

**Data availability statement:** All relevant data are within the manuscript and its Supporting Information files.

**Funding:** HN, KM, TM, ME, TK. The Japan Society for the Promotion of Science JSPS KAKENHI Grant (No. 19K09779, 21K09447, 22K09570) from the Ministry of Education, Science, and Culture of Japan (Tokyo, Japan). The funders had no role in study design, data collection and analysis, decision to publish, or preparation of the manuscript.

**Competing interests:** The authors have declared that no competing interests exist.

understood. β2GPI is important in the pathophysiology of APS, which is typically characterized by (venous and/or arterial) thrombosis and/or an adverse pregnancy outcome (e.g., unexplained pre-fetal and fetal death, pre-eclampsia (PE), and placental insufficiency with severe features) in the presence of persistent laboratory evidence of aPL [1]. The primary antigen to which aPL (i.e., lupus anticoagulants (LA), anti-cardiolipin (aCL), and anti-β2GPI) binds is β2GPI, a phospholipid-binding plasma protein [2]. Circulating aCL from patients with APS (but not from non-APS patients) required β2GPI as a co-factor for CL binding, and the effects of LA were shown to be sensitive to β2GPI in these patients [3–6]. On the other hand, circulating β2GPI levels were significantly higher in APS patients with a history of thrombosis than they were in healthy controls and patients with an autoimmune disease other than APS [7]. These findings cannot simply be explained by β2GPI being the major antigen for aPL in APS. Previous studies suggested the involvement of β2GPI in both the coagulation and complement systems [3,8,9]; however, it has not yet been defined in detail. Furthermore, although β2GPI was found to both promote and inhibit serine protease cascades [3], the mechanisms by which these cascades are regulated have not been elucidated.

β2GPI is a highly abundant soluble blood protein that is predominantly synthesized in hepatocytes. It has a circulating blood level of approximately 0.2 mg/ml, which has been shown to vary markedly between individuals in a healthy population [10–12], regardless of gender [13,14]. In spite of the amount of circulating β2GPI prior to conception, its expression is observed on the extravillous trophoblast and syncytio-trophoblast in the placenta during pregnancy [15–18]. The circulating level of β2GPI at 8 weeks of gestation (mean ± SD, 174.1 ± 40.9 µg/ml) does not significantly differ from that in non-pregnant women (174.1 ± 45.4 µg/ml) and gradually decreases over the first 36 weeks of gestation [13]. The circulating level of β2GPI is significantly lower at 36 weeks of gestation (156.7 ± 49.2 µg/ml) than at 8 weeks of gestation [13]. However, the circulating plasma volume during pregnancy peaks in the third trimester, with an increase of 42% (95% confidence interval (CI): 38–46) at 28–34 weeks of gestation and 48% (95% CI: 44–51) at 35–38 weeks of gestation [19]. Therefore, even though the total amount of β2GPI in the circulating blood may not decrease, its concentration may be reduced by increases in the circulating plasma volume. The total amount may be slightly increased by the production of trophoblast cells. In any case, β2GPI is expressed in the placenta exposed to maternal blood, even though the amount of placental β2GPI is markedly lower than circulating β2GPI, which suggests that placental β2GPI may exhibit specific and different activities from β2GPI in the circulating blood that potentially may play an important physiological role in the establishment and maintenance of pregnancy.

β2GPI consists of five domains: 4 similar canonical complement control protein (CCP) domains I-IV and one different domain (domain V) [20]. Each of the 4 domains I-IV contains 4 cysteine residues typically forming 2 disulfide bonds, while domain V consists of one extra pair of cysteines and a 19-residue hydrophobic loop that is responsible for anchoring the protein to negatively charged phospholipids [21–24]. The existence of the unique structural properties of β2GPI, due

to the C-terminal cysteine, has recently been the subject of discussion. This amino acid is part of a disulfide bridge in domain V and is exposed on the surface of the molecule. Previous studies demonstrated that β2GPI was reduced by a thioredoxin-1-dependent mechanism or by a protein disulfide isomerase-dependent mechanism and existed in two inter-convertible biochemical configurations depending on the integrity of the disulfide bonds: oxidized and reduced [7,25,26]. Eleven disulfide bonds form in oxidized-β2GPI. In reduced-β2GPI, the disulfide bonds Cys288/Cys326 in domain V and Cys32/Cys60 in domain I were shown to be individually or simultaneously broken [25,27]. The reduced state of β2GPI is directly assessed as free thiol β2GPI [28].

In recent years, there has been a growing interest in the two interconvertible biochemical configurations of β2GPI. However, the differences in physiological effects between the reduced and oxidized forms of β2GPI remain poorly under-stood. Total circulating β2GPI levels were found to be significantly higher in APS patients with a history of thrombosis than they were in healthy controls and patients with autoimmune disease other than APS, while the level of reduced-β2GPI was significantly lower [7]. This is not only for APS patients. Among patients with coronary artery disease, reduced-β2GPI levels were significantly lower in those with than in those without diabetes mellitus (DM) [29]. For reduced-β2GPI from oxidized-β2GPI distinct functions have been reported including a protective role against oxidative stress-induced vascular endothelial cell death and anti-inflammatory activities [30–35].

In the present study, we investigated the physiological function of β2GPI in trophoblast cells, with a focus on the reduced and oxidized forms of β2GPI under the hypothesis that placental β2GPI exhibits different activity to β2GPI in the circulating blood.

## Materials and Methods

### Cell culture

Human first trimester extravillous trophoblast, HTR-8/SVneo, cells were purchased from ATCC (CRL-3271™, Manassas, VA, USA) and maintained in complete RPMI 1640 medium (RPMI 1640 medium containing 10% heat-inactivated fetal bovine serum (FBS) and penicillin–streptomycin). First trimester placenta tissues (5–10 weeks of gestation) were obtained from cases of legal abortion with consent. The isolation of primary human trophoblast cells from first trimester placenta tissues was performed as previously described [36]. Primary culture cells were maintained in 1:1 Dulbecco's modified Eagle's medium and Ham's F-12 nutrient medium containing 2 mM L-glutamine, 10% heat-inactivated FBS, and penicillin–streptomycin (Nacalai Tesque, Kyoto, Japan). HepG2 liver cells were purchased from ATCC (HB-8065™) and cultured in Eagle's Minimum Essential Medium (Nacalai Tesque) with 10% heat-inactivated FBS and penicillin–streptomycin with or without 100 ng/mL of recombinant soluble fms-like tyrosine kinase-1 (sFlt-1; also known as soluble VEGF receptor 1, R&D Systems, Minneapolis, MN, USA).

### Effects of progesterone and 17β-estradiol

HTR-8/SVneo, a primary human trophoblast culture, and HepG2 cells were placed at $1 \times 10^5$ cells in 24-well plates and cultured for 24 hours together with 10 nM of 17β-estradiol (Nacalai Tesque) and different concentrations of progesterone (Nacalai Tesque, 0, 0.1, 1.0, 10, or 100 µM).

### Cell counting

Ten microliters of cell suspension were analyzed using a hemocytometer.

### Cell viability

Cell Counting Kit-8 (Dojindo, Kumamoto, Japan) using WST-8 [2- (2-methoxy-4-nitrophenyl)-3-(4-nitrophenyl)-5-(2,4-disulfophenyl)-2H-tetrazolium, monosodium salt] was used to assess cell viability according to the manufacturer's protocol.

## ELISA

ELISA kits to assess placental growth factor (PlGF) (R&D systems, Minneapolis, MN, USA), sFlt-1 (R&D systems), and total β2GPI (Abcam, Cambridge, UK) were used according to the manufacturers' protocols.

## ELISA for quantifying the percentage of reduced (free thiol) β2GPI

The percentage of reduced (free thiol) β2GPI within the cell culture supernatant and patient samples were measured as previously described [7,30,37]. In brief, free thiols were labeled with the free thiol-specific biotinylated probe, Nα-(3-maleimidylpropionyl) biocytin (MPB) (Thermo Fisher Scientific), and then added to streptavidin plates to capture MPB-labeled proteins after blocking with 2% BSA. To detect only β2GPI labeled with the MPB (and not other MPB-labeled proteins), rabbit anti-human β2GPI antibody, which directed to domain I of β2GPI (ABIN6288944, antibodies-online GmbH Aachen Germany), was incubated with MPB-labeled proteins. After washing the plates, alkaline phosphatase-conjugated goat anti-rabbit IgG was added, and the optical density was read at 405 nm following the addition of a chromogenic substrate. As a positive control, we used the same pooled serum from 10 healthy controls for all study.

## Preparation of reduced (free thiol) β2GPI

Reduced β2GPI was prepared by the following methods, which were described by Ioannou et al. [30]. The human APOH recombinant protein (β2GPI; 1 µM) (MBS2009346, MyBioSource, Inc. San Diego, CA, USA) was preincubated with thioredoxin-1 (TRX-1) (3.5 µM) activated with thioredoxin reductase (TRX-R) (10 nM) plus nicotinamide adenine dinucleotide phosphate (NADPH; 200 µM) to generate free thiols within β2GPI. HTR-8/SVneo cells were incubated with different concentrations of native and reduced (free thiol) recombinant β2GPI (0.4, 4.0, 40, and 400 nM) for 24 hours.

## Overexpression of the human *APOH* gene

The human apolipoprotein H/*APOH* gene ORF cDNA clone expression plasmid, C-GFPSpark® tag (HG11221-ACG, Sino Biological Inc., Beijing, China) or the pCMV3-C-GFPSpark® negative control vector (C-terminal GFPSpark®-tagged, CV026, Sino Biological Inc.) as a control (total 1000 ng DNA each per well in a 24-well plate) were transferred into cells using Lipofectanine® 3000 Reagent (Thermo Fisher, Tokyo, Japan) according to the manufacturer's instructions. Twenty-four hours after gene transfection, 3.5 nM of TRX-1 and 10 pM of TRX-R with 200 nM of NADPH were added into wells for the enzymatic reduction of β2GPI.

## Migration assay

The Cytoselect™ 24-well cell migration assay kit (Cell Biolabs, Inc., San Diego, CA) was used according to the manufacturer's protocol. Briefly, cells were seeded on polycarbonate membrane cell culture inserts with a pore size of 8 µm. After a 24-hour culture, culture medium was removed from the inside of the insert and the interior of the insert was then gently swabbed with the wet end of cotton-tipped swabs to remove non-migratory cells. Each insert was transferred to a clean well containing 400 µl of cell stain solution and incubated at room temperature for 10 minutes. After washing, inserts were transferred to an empty well and 200 µl of extraction solution was added and incubated for 10 minutes. OD was measured at 540 nm.

## Study approval

This study was approved by the Institutional Review Board (IRB) of Osaka University Medical School Hospital. Written informed consent was obtained from all patients for providing blood samples using a comprehensive consent form (#11111–4) and for providing first trimester placenta tissues using specific consent form (#20283(T2)). The retrospective study used blood samples and patient information collected and stored after consent was obtained using a comprehensive

consent form under IRB approval (#20283(T2)). For the blood samples in this study, instead of obtaining informed consent from each patient in accordance with the IRB, opt-out was done over the web. Samples were accessed for research purpose over between 15 November 2020 and 31 December 2020. All cases were assigned a study-specific identifier and the protected health information was redacted to the extent that the authors did not have access to information that could identify individual participants during or after data collection.

## Blood samples

Women with a singleton pregnancy who received prenatal care at the Osaka University Medical School Hospital between April 1, 2013 and December 31, 2020 and who had no underlying medical conditions were enrolled in this study. PE was defined as gestational hypertension accompanied by one or more of the following new-onset conditions at or after 20 weeks of gestation that normalized by 12 weeks postpartum, as per the International Society for the Study of Hypertension in Pregnancy [38]: proteinuria, liver involvement without any underlying diseases, progressive kidney injury, stroke, neurological complications, hematological complications, and uteroplacental dysfunction. There were no women with APS in the PE (n = 26) or control (n = 14) groups. None of the women in the control group had evidence of PE, gestational hypertension, chorioamnionitis, chronic hypertension, or a medical history that suggested they were at an increased risk of developing PE. They were taking no medications. Blood samples were obtained from patients at term before delivery.

## Statistical analysis

The experimental results are expressed as the mean ± the standard error of the mean (SEM). All data were statistically analyzed using JMP® Pro version 17.0.0 software (SAS Institute, Cary, NC, USA) and SigmaPlot® software 14.5 (Systat Software, Inc., San Jose, CA, USA). Comparisons among groups were conducted using the one-way Analysis of Variance (ANOVA) or Kruskal-Wallis ANOVA on Ranks with Shapiro–Wilk normality test and Brown–Forsythe test, followed by Student-Newman-Keuls multiple comparison test. To assess the difference between the two groups, data were analyzed using the Student's t-test with the Shapiro–Wilk normality test.

## Results

### Progesterone regulates β2GPI secretion in trophoblast cells

If placental β2GPI acts differently from β2GPI in the circulating blood, local exposure to higher levels of pregnancy-related hormones may have an impact on the effects of β2GPI in trophoblast cells. Therefore, we examined the effects of pregnancy-related hormones on β2GPI secretion in trophoblast cells using the first-trimester trophoblast cell line, HTR-8/SVneo, and primary culture first-trimester trophoblast cells.

The addition of human chorionic gonadotropin (0.1 or 100 mIU/ml) did not affect the secretion of total-β2GPI in HTR-8/SVneo cells (S1 Fig). On the other hand, 24 hours after the addition of 10 nM of 17β-estradiol ($E_2$) with progesterone (P) (1.0 or 10 µM), the total number of cells ($P < 0.001$, vs. control; Fig 1A), cell viability ($P < 0.001$, vs. control; Fig 1B) significantly increased in HTR-8/SVneo cells. Only 1 µM of P with 10 nM of $E_2$ significantly increased the secretion of total-β2GPI in HTR-8/SVneo cells ($P = 0.001$, vs. control; Fig 1C). Similarly, in primary culture trophoblast cells, the addition of 10 nM $E_2$ with P (1.0 or 10 µM) significantly increased the total number of cells ($P < 0.001$, vs. control; Fig 1E), cell viability ($P < 0.001$, vs. control; Fig 1F), and the secretion of total-β2GPI ($P < 0.001$, vs. control; Fig 1G). Following the addition of P above physiological concentrations (100 µM) with 10 nM of $E_2$ to HTR-8/SVneo and primary culture trophoblast cells, the number of trophoblast cells ($P = 0.006$; Fig 1A, $P < 0.001$; Fig 1E) and cell viability ($P = 0.002$; Fig 1B, $P < 0.001$; Fig 1F) were significantly lower than those in the control groups, while total-β2GPI levels were suppressed to the same extent as those in each control group.

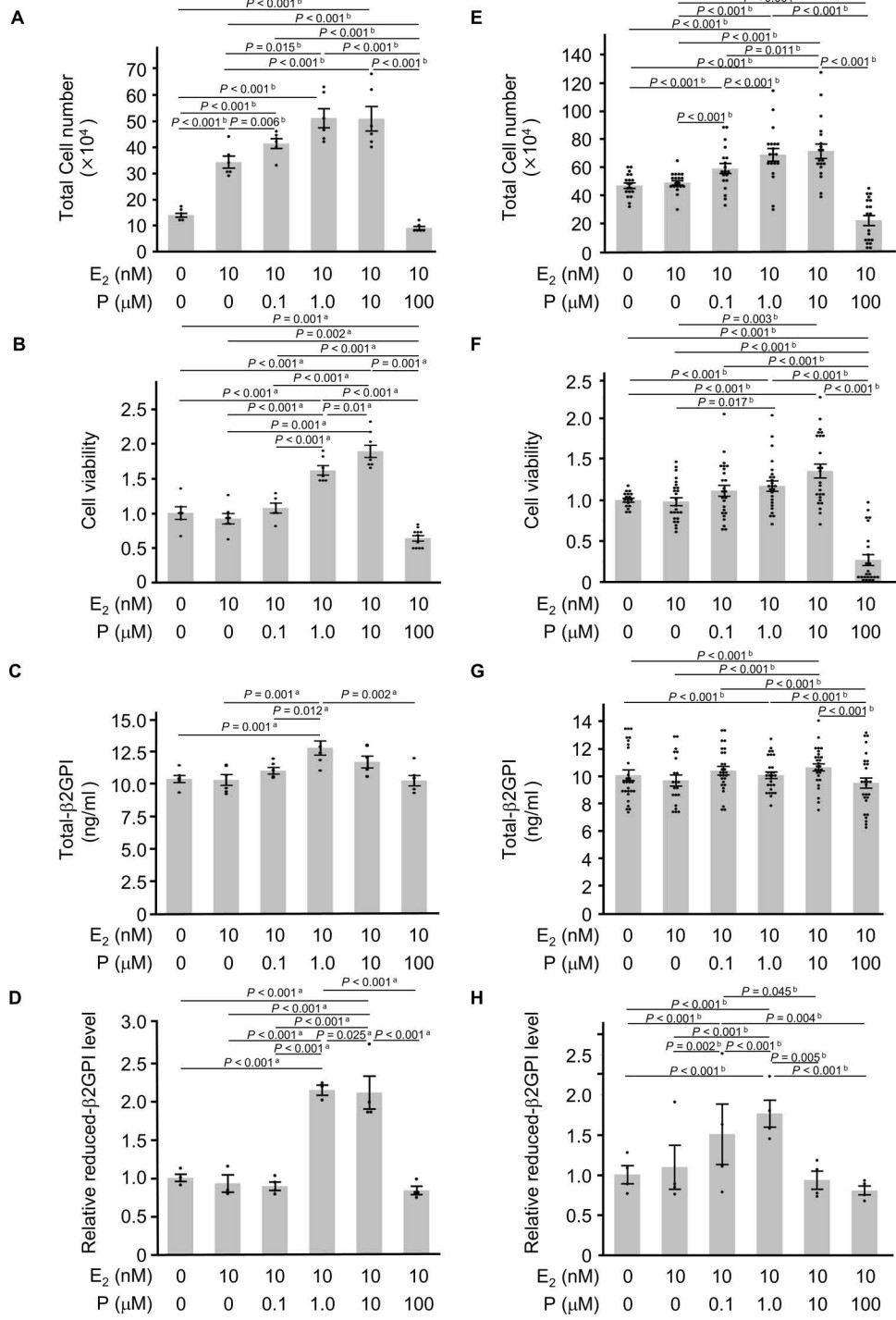

**Fig 1. Effects of progesterone on trophoblast cells.** Twenty-four hours after the treatment with 17β-estradiol (E₂) and progesterone (P), (A, E) total cell numbers (n = 6 each), (B, F) cell viability (n = 6, n = 5, respectively), (C, G) total-β2GPI levels in culture supernatants (n = 6, n = 5, respectively), and (D, H) the relative levels of reduced-β2GPI in culture supernatants (n = 4 each) were assessed in (A, B, C, D) HTR-8/SVneo and (E, F, G, H) primary culture human trophoblast (4 samples) cells. All bars represent the mean ± SEM. Dot plots for each measurement value were superimposed onto the bar graph. Data were evaluated by one-way ANOVA (a) or Kruskal-Wallis ANOVA on Ranks (b) with Shapiro–Wilk normality test and Brown–Forsythe test, followed by Student-Newman-Keuls multiple comparison test.

The secretion of reduced-β2GPI in HTR-8/SVneo cells was significantly increased by the addition of 10 nM of $E_2$ with P (1.0 or 10 μM) (2.2- or 2.1-fold, respectively, $P < 0.001$, vs. control; Fig 1D). Similarly, in primary culture cells, the addition of 10 nM of $E_2$ with 0.1 or 1.0 μM of P increased the secretion of reduced-β2GPI (1.5- or 1.8-fold, respectively, $P < 0.001$, vs. control; Fig 1H). The majority of the increase observed in total-β2GPI production following the addition of $E_2$ and P was in the reduced form in both HTR-8/SVneo and primary culture cells. HTR-8/SVneo cells were used in subsequent experiments as a human trophoblast cell model.

## Hepatocyte Response: Progesterone Modulates β2GPI Secretion

In contrast to trophoblast cells, the addition of 10 nM $E_2$ alone significantly decreased the secretion of total-β2GPI in HepG2 hepatocyte cells (−14.4%, $P = 0.009$; Fig 2). Moreover, the addition of 10 nM of $E_2$ with P (1.0, 10, or 100 μM) significantly decreased the secretion of total-β2GPI (−16.5%, $P = 0.038$, −22.8%, $P = 0.0004$, −35.9%, $P = 0.0004$, respectively, vs. the control) in a dose-dependent manner (Fig 2). However, the addition of 10 nM of $E_2$ with P (1.0, 10, or 100 μM) significantly increased the secretion of reduced-β2GPI ($P < 0.001$, vs. the control; Fig 2).

## β2GPI synthesized in hepatocytes differs from placental β2GPI

To examine the effects of endogenous β2GPI, *APOH* gene expression plasmid DNA (pCMV3-*APOH*-GFPSpark) or control plasmid DNA (pCMV3-control-GFPSpark) was transferred into HTR-8/SVneo (Fig 3A) and HepG2 (Fig 3B) cells. The secretion of total-β2GPI significantly increased in the *APOH*-transferred groups ($P = 0.006$, $P = 0.002$, without or with TRX-1 treatment group, respectively. vs. control plasmid DNA-transferred group with no TRX treatment, $P < 0.001$, vs. control plasmid DNA-transferred group with TRX-1 treatment group; Fig 3A). In the *APOH*-transferred groups, the secretion of reduced-β2GPI was similar regardless of the TRX-1 treatment (Fig 3A). This result suggests that the majority of endogenous β2GPI secreted in trophoblast cells was the reduced form.

In contrast, the overexpression of *APOH* cDNA significantly increased total-β2GPI secretion in HepG2 hepatocyte cells ($P = 0.0136$, Student's *t*-test, Fig 3B), whereas no significant difference was noted in the level of reduced-β2GPI between the β2GPI-transferred and control gene-transferred groups (Fig 3B). These results suggest that, in contrast to placental β2GPI, β2GPI synthesized in hepatocytes was predominantly the oxidized form.

## Oxidized-β2GPI, but not reduced (free thiol)-β2GPI, stimulates the secretion of sFlt-1 in trophoblast cells, which may result in the excessive production of oxidized-β2GPI in hepatocytes

The induction and inhibition of angiogenesis by β2GPI have been reported and were found to be dependent on its conformation and concentration [31,39–41]. In a syngeneic, immunocompetent mouse melanoma model, β2GPI-derived peptides increased sFlt-1, which is an anti-angiogenic factor, and suppressed melanoma tumor growth and vascular endothelial cell migration [42]. Soluble Flt-1 acts as a potent antagonist of VEGF and PlGF by binding these molecules in the circulation. The imbalance between PlGF and sFlt-1 has been implicated in the pathophysiology of PE and is used clinically as a biomarker [43]. On the other hand, even in normal pregnancy in mice and women, sFlt-1 is expressed in placental trophoblast cells from the early stages of pregnancy [44,45]. In the present study, the secretion of PlGF in trophoblast cells was not affected by the overexpression of the *APOH* gene or the TRX-1 treatment (Fig 4A). However, a significant reduction (approximately 60%) was observed in sFlt-1 secretion in the *APOH* gene overexpression groups with or without the TRX-1 treatment ($P = 0.001$, $P < 0.001$, without or with TRX-1 treatment group, respectively. vs. control plasmid DNA-transferred group with no TRX treatment, $P = 0.004$, $P = 0.003$, vs. control plasmid DNA-transferred group with TRX-1 treatment group; Fig 4A).

To investigate whether the oxidized and reduced forms of β2GPI exerted different effects on trophoblast cells, recombinant β2GPI (rβ2GPI) was added to trophoblast cells. The absorbance (optical density; OD = 405 nm) of native-rβ2GPI was not changed by MPB labeling and was approximately 2.5-fold higher following the addition of TRX-1 and TRX-R

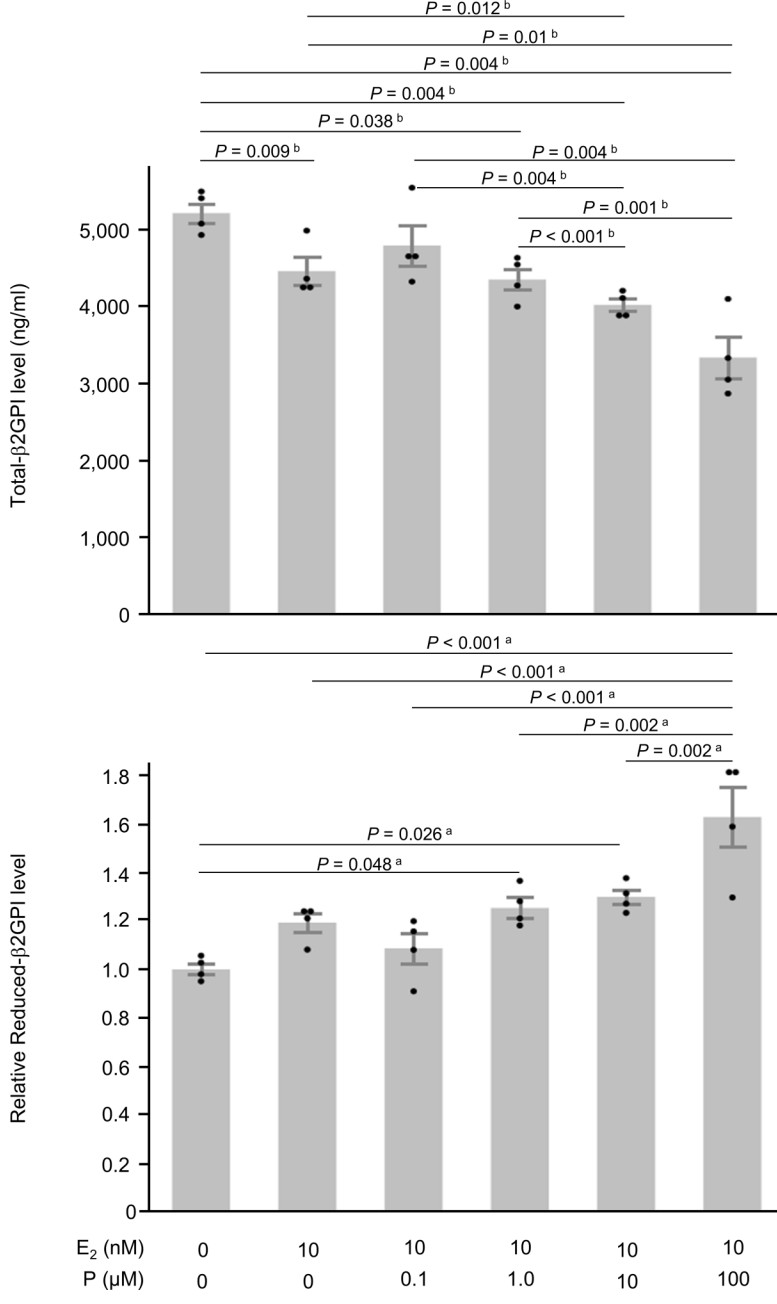

**Fig 2. Effects of progesterone on hepatocyte cells.** Twenty-four hours after the treatment with 17β-estradiol ($E_2$) and progesterone (P), total-β2GPI levels (n = 4) and the relative levels of reduced-β2GPI (n = 4) in culture supernatants (n = 4) were assessed in HepG2 cells. All bars represent the mean ± SEM. Dot plots for each measurement value were superimposed onto the bar graph. Data were evaluated by one-way ANOVA (a) or Kruskal-Wallis ANOVA on Ranks (b) with Shapiro–Wilk normality test and Brown–Forsythe test, followed by Student-Newman-Keuls multiple comparison test.

(S2 Fig). This result suggests that the majority of native-rβ2GPI was oxidized. The addition of different concentrations (0, 0.4, 4.0, 40, or 400 nM) of native-rβ2GPI (as oxidized-β2GPI) or TRX-1-treated rβ2GPI (as reduced-β2GPI) did not affect cell proliferation (S3 Fig). Only the addition of 40 or 400 nM native-rβ2GPI (as oxidized-β2GPI) significantly decreased the secretion of PlGF ($P = 0.027$, $P = 0.032$, vs. 4 nM native- rβ2GPI; Fig 4B). The addition of TRX-1-treated rβ2GPI

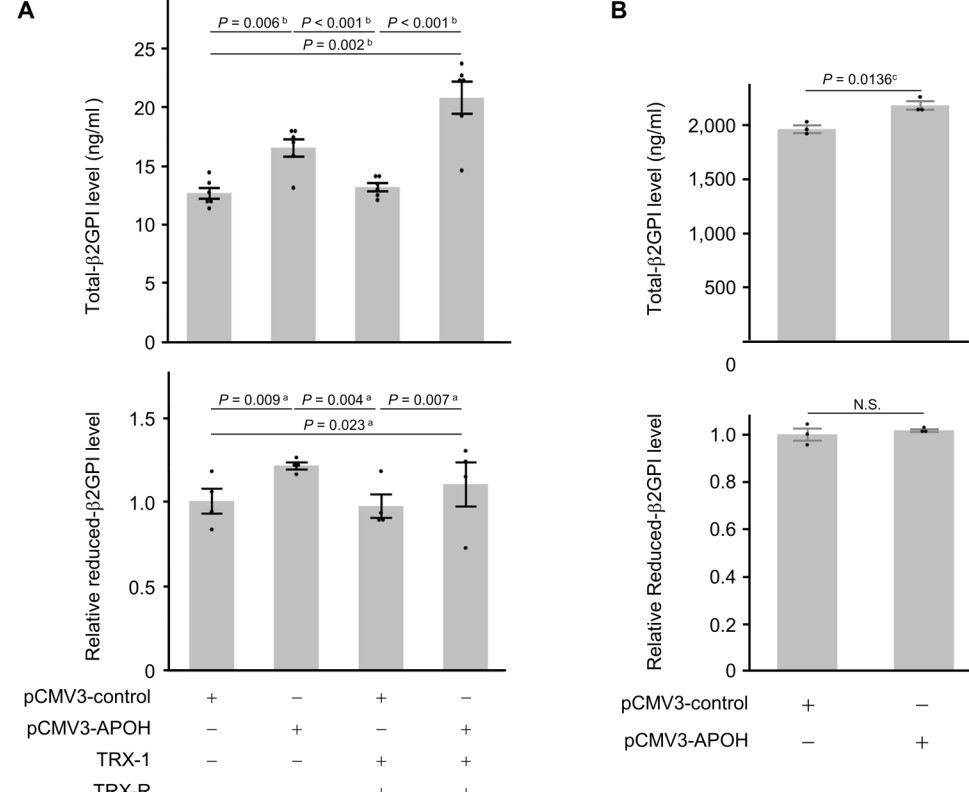

**Fig 3. Effects of endogenous β2GPI.** To investigate the effects of β2GPI on trophoblast and hepatocyte cells, *APOH* gene expression plasmid DNA (pCMV3-*APOH*) or control plasmid DNA (pCMV3-control) was transferred into (A) HTR-8/SVneo or (B) HepG2 cells and thioredoxin-1 (TRX-1) and thioredoxin reductase (TRX-R) were then added to generate free thiols within β2GPI. Total-β2GPI levels in culture supernatants (n = 6) and the relative levels of reduced (free thiol) β2GPI in culture supernatants (n = 6) were assessed. All bars represent the mean ± SEM. Dot plots for each measurement value were superimposed onto the bar graph. Data were evaluated by one-way ANOVA (a) or Kruskal-Wallis ANOVA on Ranks (b) with Shapiro–Wilk normality test and Brown–Forsythe test, followed by Student-Newman-Keuls multiple comparison test, or the Shapiro–Wilk normality test and Student's *t*-test (c).

(as reduced-β2GPI) did not change sFlt-1 secretion in trophoblast cells, whereas the addition of native-rβ2GPI (as oxidized-β2GPI, 4.0, 40, or 400 nM) significantly increased sFlt-1 secretion in a dose-dependent manner (*P* < 0.01, *P* < 0.001, *P* = 0.025, respectively, vs. the negative control; Fig 4B).

The addition of native-rβ2GPI at a concentration of 400 nM significantly suppressed HTR-8/SVneo trophoblast cell migration, with a decrease of approximately 80% in cell numbers (*P* < 0.001, vs. control group; Fig 4C). No significant differences were noted between the TRX-I treated-rβ2GPI (400 nM) and control groups.

If excess placental sFlt-1 due to oxidized-β2GPI increases circulating levels of sFlt-1, does an increase in circulating sFlt-1 affect the secretion and redox status of β2GPI in hepatocytes? While no significant differences were observed between the addition of 1 or 10 ng/ml of sFlt-1 and the control groups, the addition of 100 ng/ml of sFlt-1 significantly increased total-β2GPI levels (*P* = 0.018, vs. control group; Fig 4D) and significantly decreased reduced-β2GPI levels (*P* = 0.036, vs. control group; Fig 4D) in HepG2 hepatocyte cells.

## Women with PE have an elevated level of circulating oxidized-β2GPI

Oxidized-β2GPI suppressed trophoblast migration and increased sFlt-1 secretion in trophoblast cells. If oxidized-β2GPI also leads to excess circulating sFlt-1, it may trigger a negative spiral in which its secretion in hepatocytes is increased,

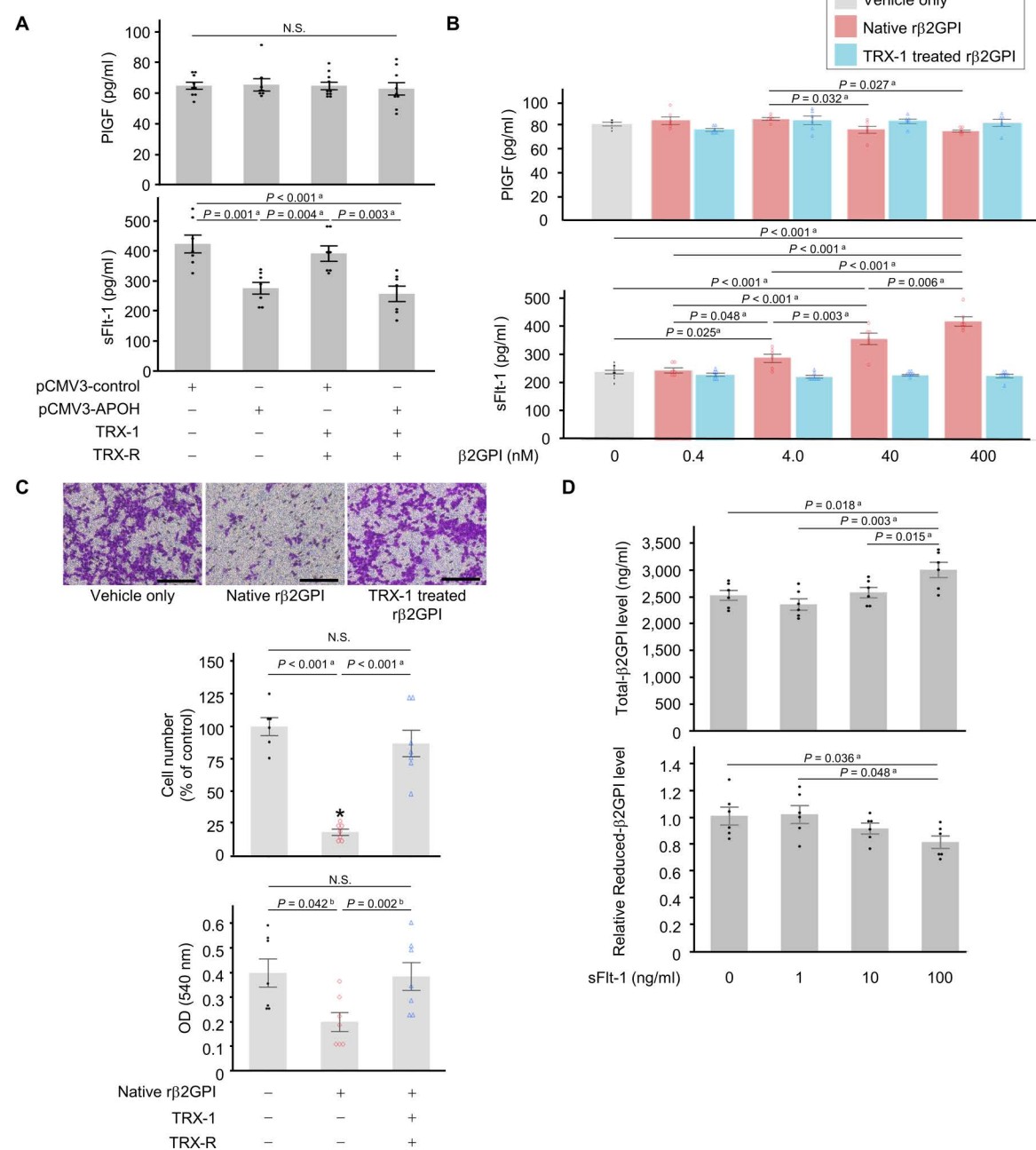

**Fig 4. Impact of the β2GPI redox status on trophoblast cells.** (A) The secretion of placental growth factor (PlGF) (n = 9) and soluble fms-like tyrosine kinase-1 (sFlt-1) (n = 7) were assessed in HTR-8/SVneo trophoblast cells following the transfer of *APOH* gene expression plasmid DNA (pCMV3-APOH) or control plasmid DNA (pCMV3-control) and subsequent treatment with thioredoxin-1 (TRX-1) and thioredoxin reductase (TRX-R). (B) After incubation with different concentrations (0, 0.4, 4.0, 40, or 400 nM) of native recombinant β2GPI (native rβ2GPI), most of which was in its oxidized form (See S2 Fig), or TRX-1-treated rβ2GPI, which generates free thiols within β2GPI (reduced-β2GPI), for 24 hours, the secretion of PlGF (n = 6) and sFlt-1 (n = 6) in HTR-8/SVneo cells were assessed. (C) A migration assay was performed 24 hours after the addition of native or TRX-1-treated recombinant β2GPI (rβ2GPI) (400 nM). Migratory cells on the bottom of the membrane were stained, and cell numbers were assessed and quantified at OD 540 nm after extraction. (D) The levels of total-β2GPI and reduced-β2GPI in culture supernatants were assessed in HepG2 liver cells 24 hours after the addition of sFlt-1 (0, 1, 10, and 100 ng/ml). All bars represent the mean ± SEM. Dot plots for each measurement value were superimposed onto the bar graph. Data were evaluated by one-way ANOVA (a) or Kruskal-Wallis ANOVA on Ranks (b) with Shapiro–Wilk normality test and Brown–Forsythe test, followed by Student-Newman-Keuls multiple comparison test. Scale bars: 200 μm.

resulting in a higher level of circulating oxidized-β2GPI that may induce trophoblast cells to secrete more sFlt-1. Therefore, circulating total-β2GPI levels and the redox status of β2GPI were examined in women with PE. No significant differences were observed in background characteristics between the PE and control groups (Table 1). Although total-β2GPI levels were significantly higher in the PE group (mean, 95% CI, 206.3 μg/ml, 188.5–227.1 μg/ml) than in the control group (161.9 μg/ml, 133.6–190.3 μg/ml, $P$ = 0.0148, Student's $t$-test, Fig 5A), reduced-β2GPI (free thiol β2GPI) levels were significantly lower in the PE group (mean, 95% CI, 64.4%, 56.0–72.7%) than in the control group (85.8%, 74.9–96.7%, $P$ = 0.0031, Student's $t$-test, Fig 5B). These results suggest that circulating oxidized-β2GPI levels increased in women with PE.

## Discussion

At approximately 9 weeks of gestation, placental trophoblast cells take over from the corpus luteum as the primary source of P production [46]. Therefore, trophoblast cells themselves may be exposed to a higher concentration of P than that in the circulating blood. Our initial hypothesis was that local exposure to high concentrations of P in the placenta may temporarily and reversibly affect the function of β2GPI in the circulating blood. However, this hypothesis was proven incorrect. Placental β2GPI was originally distinct from hepatocyte β2GPI. The addition of 10 nM of $E_2$ with P (1.0 or 10 μM), which represents a pregnancy model condition, significantly increased total-β2GPI, which was mostly the reduced form, in trophoblast cells (Fig 1). Conversely, the addition of P suppressed the secretion of total-β2GPI in HepG2 hepatocyte cells in a dose-dependent manner (Fig 2), although the majority of β2GPI from HepG2 cells was the oxidized form (Fig 3B). These results suggest that in HepG2 cells, P suppresses the secretion of β2GPI and simultaneously protects against post-translational modifications to β2GPI, which is the process of the oxidation of free cysteine thiols in domain V. In contrast, placental β2GPI was mostly the reduced form, irrespective of the presence of P (Figs 1). Collectively, the present results indicate that β2GPI from trophoblast cells, which were mostly the reduced form, at least *in vitro*, did not affect trophoblast cell migration from that in the control group and acted in an inhibitory manner on sFlt-1 production in trophoblast cells. When β2GPI is oxidized, it could inhibit trophoblast cell migration and induce an increase in sFlt-1 production in trophoblast cells. Placental β2GPI (reduced form of β2GPI) may regulate the placenta to avoid an excess of placental

**Table 1. Patients characteristics.**

|  | Control (n = 14) | Preeclampsia (n = 26) | *P*-value |
|---|---|---|---|
| Age, years | 36.1 (32.7-39.6) | 35.8 (33.3-38.3) | N.S. [a] |
| Race and ethnicity Asian, n (%) | 14 (100) | 26 (100) | – |
| Primiparity, n (%) | 9 (64.3) | 21 (80.7) | N.S. [b] |
| Body mass index (kg/ m²) | 20.8 (19.2-22.3) | 20.7 (19.6-21.9) | N.S. [a] |
| Smoking, n (%) | 0 | 0 | – |
| ART pregnancy, n (%) | 5 (35.7) | 6 (23.1) | N.S. [b] |
| Pre-gestational DM, n (%) | 0 | 0 | – |
| GDM, n (%) | 0 | 0 | – |
| Kidney disease, n (%) | 0 | 3 (11.5) | N.S. [b] |
| Thyroid disease, n (%) | 0 | 3 (11.5) | N.S. [b] |
| Previous HDP, n (%) | 0 | 1 (3.8) | N.S. [b] |
| GA at onset of preeclampsia | – | 38.3 (37.2-39.5) | – |
| GA at delivery | 38w6d (38w1d-40w0d)¤ | 38w4d (37w6d-39w1d) | N.S. [a] |

Data are presented as mean (95% confidence interval) or number of patients (%).

ART, assisted reproductive technology; GDM, gestational diabetes mellitus; DM, diabetes mellitus; HDP, hypertensive disorders of pregnancy; GA, gestational age; N.S., not significant.

[a]Student's t-test, [b] 2-sided P-value by Fisher's exact test.

**A**

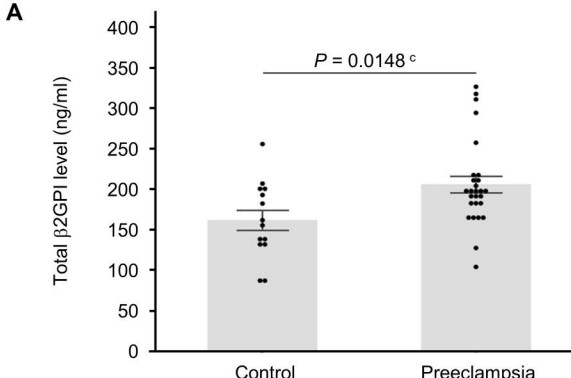

**B**

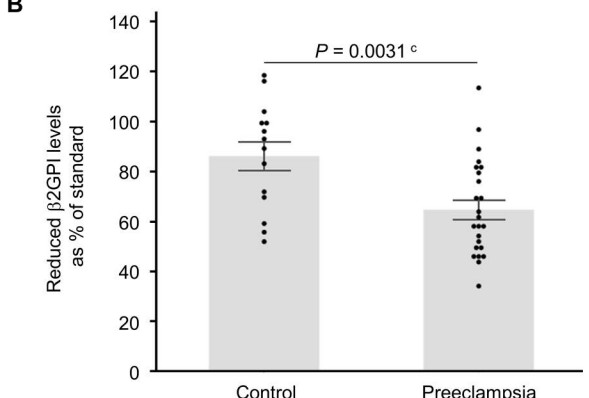

**Fig 5. Circulating levels of total- and reduced- (free thiol) β2GPI in women with pre-eclampsia (PE).** Peripheral blood samples were collected at term before delivery. Total and reduced (free thiol) β2GPI levels in circulating blood samples were assessed in the control (n = 14) and PE (n = 26) groups. All bars represent the mean ± SEM. Dot plots for each measurement value were superimposed onto the bar graph. Data were evaluated by the Shapiro–Wilk normality test and Student's *t*-test (*$P < 0.05$, **$P < 0.01$).

sFlt-1. In contrast, increased oxidative β2GPI under oxidative stress may lead to an excess of placental sFlt-1, which may contribute to the pathogenesis of PE.

PE is a serious pregnancy complication with significant maternal and fetal morbidities and mortalities, which are characterized by new-onset hypertension and often proteinuria in the mother [47]. If the fetus and placenta are not delivered, it can progress to multi-organ dysfunction, including hepatic, renal, and cerebral diseases. PE is considered to originate in the placenta [48]. A crucial factor in the development of PE is inadequate placental invasion, which leads to an excess of placental sFlt-1 [44,49–51] and soluble endoglin [52], which are anti-angiogenic factors that enter the maternal circulation and contribute to endothelial dysfunction. The concept that PE is a multisystemic syndrome is recognized in both research and clinical care [53]. Placental and systemic oxidative stress is regarded as one of the key aspects in the pathogenesis of PE [54]. In the placenta, poor maternal uterine arterial remodeling during placentation has been suggested to cause placental ischemia-reperfusion injury and oxidative stress due to the complications associated with inappropriate pressure on blood flow rather than chronic hypoxia itself [55]. Maternal comorbidities, such as DM and obesity, are known risk factors for PE [48], as they are believed to increase systemic oxidative stress [56]. Patients diagnosed with type 2 DM had significantly higher levels of total-β2GPI [57] and lower levels of reduced-β2GPI [58] in the circulating blood than a healthy population, suggesting an increase in oxidized-β2GPI.

Very recently, placenta-tropic lipid nanoparticles (LNPs) using a mechanism based on absorption with endogenous β2GPI have been reported [59]. The study demonstrated that the uptake of LNPs in the liver was significantly increased in β2GPI knockdown mice following intravenous siRNA administration, while it was reduced in the placenta [59]. Furthermore, a significant increase in blood β2GPI levels was observed in inflammation-induced PE model mice, and these mice showed a higher uptake of LNPs in the liver but not in the placenta. These results suggest the possibility of differential actions of β2GPI in the placenta and in the liver. The present study indicates that the oxidation of β2GPI and excess placental sFlt-1 trigger a negative spiral in which the secretion of oxidized-β2GPI in hepatocytes is increased, resulting in a higher level of circulating oxidized-β2GPI that induces trophoblast cells to secrete more sFlt-1. According to the International Society for the Study of Hypertension in Pregnancy (ISSHP) classification, the HELLP syndrome (Hemolysis, Elevated Liver enzymes, Low platelets) is a serious manifestation of PE and not a separate disorder [60]. Circulating sFlt-1 levels were previously shown to be higher in PE associated with HELLP syndrome than in severe PE, which was diagnosed according to the 2013 American College of Obstetricians and Gynecologists (ACOG) guidelines on hypertension in pregnancy [47,61], while another group demonstrated that women with PE associated with HELLP syndrome had a higher median level of sFlt-1 than women with PE, which was not significantly different due to high variability [62]. If β2GPI oxidation affects sFlt-1 secretion in the placenta and excessive sFlt-1 affects circulating β2GPI levels, β2GPI may be involved the pathophysiology of HELLP syndrome because together with C-reactive protein and thrombomodulin, β2GPI is the only protein with a dual function of up- and down-regulating the complement and coagulation systems in response to external stimuli [63]. Moreover, women with acute fatty liver of pregnancy (AFLP), which is an obstetric emergency characterized by maternal liver dysfunction and/or failure that may lead to maternal and fetal complications, including death, showed significantly higher sFlt-1 levels than women with HELLP syndrome [64]. A correlation has been reported between the severity of AFLP by the Swansea criteria and sFlt-1 levels [64]. The pathogenesis of HELLP syndrome and AFLP may involve a negative spiral due to the oxidation of β2GPI and excess placental sFlt-1. However, further studies are needed to confirm this.

In conclusion, it is important to note that placental β2GPI may differ from β2GPI synthesized in hepatocytes. Placental β2GPI, which is predominantly the reduced form, has a different composition from hepatocyte β2GPI, where β2GPI in the circulating blood is predominantly synthesized in hepatocytes. From a physiological perspective, the expression of placental β2GPI may play a role in protecting the placenta from oxidative stress and maintaining homeostasis in the pregnancy environment.

## Supporting information

**S1 Fig. Effects of human chorionic gonadotropin (hCG) on trophoblast cells.** An assessment of (A) the total cell number, (B) CCK-8 assay, and (C) total-β2GPI levels in culture supernatants was performed 24 hours after the addition of hCG (0.1 and 100 mIU/ml). All bars represent the mean ± SEM. Dot plots for each measurement value were superimposed onto the bar graph. Data were evaluated by one-way ANOVA with Shapiro–Wilk normality test and Brown–Forsythe test, followed by Student-Newman-Keuls multiple comparison test.
(TIF)

**S2 Fig. Characteristic of native recombinant β2GPI.** After labeling with the free thiol-specific biotinylated probe, Nα-(3-maleimidylpropionyl) biocytin (MPB), recombinant β2GPI with or without the pretreatment of thioredoxin-1 (TRX-1) and thioredoxin reductase (TRX-R) were incubated on a streptavidin plate and probed with an anti-β2GPI antibody. After washing plates, alkaline phosphatase-conjugated goat anti-rabbit IgG was added, and optical density was read at 405 nm after the addition of a chromogenic substrate. Data were evaluated by one-way ANOVA with Shapiro–Wilk normality test and Brown–Forsythe test, followed by Student-Newman-Keuls multiple comparison test.
(TIF)

**S3 Fig. Effects of exogenous β2GPI on trophoblast cells.** HTR-8/SVneo trophoblast cells were incubated for 24 hours with different concentrations (0, 0.4, 4.0, 40, or 400 nM) of native recombinant β2GPI (rβ2GPI), most of which was in its oxidized form (See S2 Fig), or TRX-1-treated rβ2GPI, which generates free thiols within β2GPI (reduced-β2GPI). (A) Total cell number (n = 6) and (B) cell viability (n = 6) were assessed. All bars represent the mean ± SEM. Dot plots for each measurement value were superimposed onto the bar graph. Data were evaluated by one-way ANOVA with Shapiro–Wilk normality test and Brown–Forsythe test, followed by Student-Newman-Keuls multiple comparison test.
(TIF)

**S1 Data.**
(XLSX)

## Acknowledgments

The authors thank Dr Takeshi Hisamatsu (Hisamatsu Maternity Clinic) for organizing the clinical samples. We extend our gratitude to all patients who consented to provide specimens.

## Author contributions

**Conceptualization:** Kazunobu Yagi, Hitomi Nakamura.

**Data curation:** Kazunobu Yagi, Reina Komatsu, Hitomi Nakamura, Kazuya Mimura, Masayuki Endo, Takuji Tomimatsu, Tadashi Kimura.

**Formal analysis:** Kazunobu Yagi, Reina Komatsu, Hitomi Nakamura.

**Funding acquisition:** Hitomi Nakamura, Kazuya Mimura, Masayuki Endo, Takuji Tomimatsu, Tadashi Kimura.

**Investigation:** Kazunobu Yagi, Reina Komatsu, Hitomi Nakamura, Kazuya Mimura, Masayuki Endo, Takuji Tomimatsu, Tadashi Kimura.

**Methodology:** Kazunobu Yagi, Reina Komatsu, Hitomi Nakamura, Kazuya Mimura, Masayuki Endo, Takuji Tomimatsu, Tadashi Kimura.

**Project administration:** Hitomi Nakamura.

**Resources:** Hitomi Nakamura, Kazuya Mimura, Masayuki Endo, Takuji Tomimatsu, Tadashi Kimura.

**Supervision:** Hitomi Nakamura, Tadashi Kimura.

**Validation:** Kazunobu Yagi, Reina Komatsu.

**Visualization:** Kazunobu Yagi, Reina Komatsu, Hitomi Nakamura.

**Writing – original draft:** Kazunobu Yagi, Hitomi Nakamura.

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
