## [Decision Letter · Decision Letter 0]

17 Dec 2024

PONE-D-24-44937Beyond antibodies: Beta-2 glycoprotein I as the unsung guardian of pregnancyPLOS ONE

Dear Dr. Nakamura,

Thank you for submitting your manuscript to PLOS ONE. After careful consideration, we feel that it has merit but does not fully meet PLOS ONE’s publication criteria as it currently stands. Therefore, we invite you to submit a revised version of the manuscript that addresses the points raised during the review process.

While the reviewers provided differing perspectives on the extent of the revisions required, we value the constructive feedback from both. We believe that addressing the concerns raised by Reviewer 2, in addition to incorporating Reviewer 1 suggestions, will significantly enhance the manuscript and ensure its suitability for our journal.

We look forward to receiving your revised manuscript.

Kind regards,

María Teresa Llinás

Academic Editor

PLOS ONE

“HN, KM, TM, ME, TK.

The Japan Society for the Promotion of Science JSPS KAKENHI Grant (No. 19K09779, 21K09447, 22K09570) from the Ministry of Education, Science, and Culture of Japan (Tokyo, Japan).”

“The authors thank Dr Takeshi Hisamatsu (Hisamatsu Maternity Clinic) for organizing the clinical samples. We extend our gratitude to all patients who consented to provide specimens. This work was supported in part by the Japan Society for the Promotion of Science JSPS KAKENHI Grant (No. 19K09779, 21K09447, 22K09570) from the Ministry of Education, Science, and Culture of Japan (Tokyo, Japan).”

“HN, KM, TM, ME, TK.

The Japan Society for the Promotion of Science JSPS KAKENHI Grant (No. 19K09779, 21K09447, 22K09570) from the Ministry of Education, Science, and Culture of Japan (Tokyo, Japan).”

Reviewers' comments:

Reviewer's Responses to Questions

**Comments to the Author**

1. Is the manuscript technically sound, and do the data support the conclusions?

Reviewer #1: Partly

Reviewer #2: Yes

2. Has the statistical analysis been performed appropriately and rigorously? 

Reviewer #1: No

Reviewer #2: Yes

3. Have the authors made all data underlying the findings in their manuscript fully available?

Reviewer #1: Yes

Reviewer #2: Yes

4. Is the manuscript presented in an intelligible fashion and written in standard English?

Reviewer #1: Yes

Reviewer #2: Yes

5. Review Comments to the Author

Reviewer #1: A manuscript by Nakamura and co-authors is potentially interesting. It brings a new perspective to the involvement of β2GPI in trophoblast cell function. However, the sudy has some major issues that need to be addressed.

- No details have been given for the Ethical approval for first trimester placental tissue, but only for blood samples. Please give details on who issued the approval, and the approval number for the tissue.

- Materials and Methods section needs more detail on the procedures. For example, labeling of proteins was not described well. Were these proteins from cell culture supernatants? Because the line 153 says “the same pooled serum was used as positive control”. Same to what? It is unclear.

- Only the method for cell viability was given (CCK8), and none for cell number determination, Yet, these data are presented in the Figure 1.

- Another important issue refers to the proliferation of primary trophoblast. In usual cell culture settings, cytotrophoblast does not proliferate, but rather differentiate and fuse in syncitium. Thus, it is unclear how the authors observed the proliferationin primary trophoblast cells.

- Students t-test is not apropriate for multiple compairsons. The results should be analyzed by One-Way ANOVA, followed by appropriate post-hoc test. How were the significances obtained?The authors talk about the dose -dependence (line 263). However, the statistical differences between the treatment groups is not shown, only compared to the control.

Other remarks:

In the Abstract, it is not clearly written what was investigated in the study. A sentence od two should be added to explain what was investigated and why, before the results.

Line 111 – extravillous trophoblast cell line

Line 158 and 159 – give the full names before introducing abbrevations TRX-1 and TRX-R, and briefly explain why they are added.

Line 132 – The title is not correct, no cell proliferation method is described. It is also gramatically not correct.

To what section do lines 389-394 belong? Should they be before Figure 5 lengend? This way this paragraph stands unlinked to other text.

Reviewer #2: This article in the abstract and introduction sets the scene why this research was undertaken which is easy to follow. The researchers may also wish to briefly include a few lines either in their discussion or introduction the recent online publication in Nature 11 December 2024 by Kelsey L Swingle et al Placenta-tropic VEGF mRNA lipid nanoparticles ameliorate murine pre-eclampsia, which found that "an endogenous targeting mechanism based on β2-glycoprotein I adsorption that enables LNP delivery to the placenta." This seems to further support the hypothesis of this paper that beta 2 glycoprotein I plays an important biological role in placenta biology.

The methods and ethics and statistical methodology are sound

Results figures and tables are easy to follow and have a logical sequence

My only concern is with line 281-282 (an approximately 5 fold increase.. Fig 3A). When I look at Figure 3A the increase is much less than 5 fold, going from approximately 12.5 ng/ml of total B2GPI in control to 20ng/ml of the TRXI-TRXR treated and to 16ng/ml total B2GPI for the pCMV3-APOH without TRX1-TRXR treatment.

The discussion is easy to follow and ties the results together nicely.

6. PLOS authors have the option to publish the peer review history of their article (what does this mean? ). If published, this will include your full peer review and any attached files.

**Do you want your identity to be public for this peer review?** For information about this choice, including consent withdrawal, please see our Privacy Policy .

Reviewer #1: No

Reviewer #2: **Yes: ** Bill Giannakopoulos

---

## [Author Response · Author response to Decision Letter 1]

14 Feb 2025

and

We have read these 2 PLOS ONE’s style templates and have also reviewed the Submission Guidelines at

https://journals.plos.org/plosone/s/submission-guidelines.

We reported exact P-values for all values greater than or equal to 0.001. We have checked that the manuscript conforms to PLOS ONE style requirements, including file naming.

“HN, KM, TM, ME, TK.

The Japan Society for the Promotion of Science JSPS KAKENHI Grant (No. 19K09779, 21K09447, 22K09570) from the Ministry of Education, Science, and Culture of Japan (Tokyo, Japan).”

The funders had no role. In the cover letter we state that "The funders had no role in study design, data collection and analysis, decision to publish, or preparation of the manuscript." Could you please change the condition on the web page?

“The authors thank Dr Takeshi Hisamatsu (Hisamatsu Maternity Clinic) for organizing the clinical samples. We extend our gratitude to all patients who consented to provide specimens. This work was supported in part by the Japan Society for the Promotion of Science JSPS KAKENHI Grant (No. 19K09779, 21K09447, 22K09570) from the Ministry of Education, Science, and Culture of Japan (Tokyo, Japan).”

“HN, KM, TM, ME, TK.

The Japan Society for the Promotion of Science JSPS KAKENHI Grant (No. 19K09779, 21K09447, 22K09570) from the Ministry of Education, Science, and Culture of Japan (Tokyo, Japan).”

(Acknowledgments, line 500-502) We have removed “This work was supported in part by the Japan Society for the Promotion of Science JSPS KAKENHI Grant (No. 19K09779, 21K09447, 22K09570) from the Ministry of Education, Science, and Culture of Japan (Tokyo, Japan).”

We share the “minimal data set” for our submission.

Reviewer #1:

A manuscript by Nakamura and co-authors is potentially interesting. It brings a new perspective to the involvement of β2GPI in trophoblast cell function. However, the sudy has some major issues that need to be addressed.

- No details have been given for the Ethical approval for first trimester placental tissue, but only for blood samples. Please give details on who issued the approval, and the approval number for the tissue.

We added details for first trimester placenta tissues as below (the changes are highlighted in yellow colour);

(line 185, Study approval) This study was approved by the Institutional Review Board (IRB) of Osaka University Medical School Hospital. Written informed consent was obtained from all patients for providing blood samples using a comprehensive consent form (#11111-4) and for providing first trimester placenta tissues using specific consent form (#20283(T2)). The retrospective study used blood samples and patient information collected and stored after consent was obtained using a comprehensive consent form under IRB approval (#20283(T2)). For the blood samples in this study instead of obtaining informed consent from each patient in accordance with the IRB, opt-out was done over the web.

- Materials and Methods section needs more detail on the procedures. For example, labeling of proteins was not described well. Were these proteins from cell culture supernatants? Because the line 153 says “the same pooled serum was used as positive control”. Same to what? It is unclear.

We have changed as below;

(Line 150 in the revised manuscript with changes) The percentage of reduced (free thiol) beta2GPI within the cell culture supernatant and patient samples were measured as previously described…

(Line 161 in the revised manuscript with changes) As a positive control, we used the same pooled serum from 10 healthy controls for all of the study.

- Only the method for cell viability was given (CCK8), and none for cell number determination, Yet, these data are presented in the Figure 1.

We have added the section “cell counting” as below;

(Line 135 in the revised manuscript with changes)

Cell counting

Ten microliters of cell suspension were analyzed using a hemocytometer.

- Another important issue refers to the proliferation of primary trophoblast. In usual cell culture settings, cytotrophoblast does not proliferate, but rather differentiate and fuse in syncitium. Thus, it is unclear how the authors observed the proliferationin primary trophoblast cells.

We have changed our manuscript to mention cell number and CCK-8 assay data instead of using the term "proliferation".

- Students t-test is not apropriate for multiple compairsons. The results should be analyzed by One-Way ANOVA, followed by appropriate post-hoc test. How were the significances obtained?The authors talk about the dose -dependence (line 263). However, the statistical differences between the treatment groups is not shown, only compared to the control.

We have re-analysed data and made some changes in our manuscript as below;

(line 226 in Statistical analysis in Materials and Methods)

“Comparisons among groups were conducted using a one-way ANOVA or Kruskal-Wallis ANOVA on Ranks with Shapiro–Wilk normality test and Brown–Forsythe test, followed by Student-Newman-Keuls multiple comparison test. To assess the difference between the two groups, data were analyzed using the Student’s t-test or Wilcoxon’s rank-sum test with the Shapiro–Wilk normality test.”

Other remarks:

In the Abstract, it is not clearly written what was investigated in the study. A sentence od two should be added to explain what was investigated and why, before the results.

We have added one sentence before explaining our results as below;

(Abstract) “The physiological function of beta 2 glycoprotein I (b2GPI) itself is not well understood, other than that it is a primary antigen to anti-phospholipid antibodies in the autoimmune disease antiphospholipid syndrome. b2GPI is a soluble blood protein that is predominantly synthesized in hepatocytes. Why is the expression of b2GPI observed in the placenta despite its abundance in the circulating blood of healthy individuals? Does the placenta produce a specific-acting b2GPI?

b2GPI was recently shown to adopt two interconvertible biochemical confirmations based on the integrity of disulfide bonds: oxidized and reduced. The present study investigates the physiological function of b2GPI in trophoblast cells, with a focus on the reduced and oxidized forms of b2GPI under the hypothesis that placental b2GPI has a different activity from circulating b2GPI. Endogenous b2GPI secretion in trophoblast cells were predominantly in the reduced form, while those in HepG2 liver cells were mainly in the oxidized form. Progesterone increased reduced-b2GPI in both the trophoblast and liver cells. Oxidized-b2GPI significantly inhibited trophoblast cell migration and increased placental soluble fms-like tyrosine kinase-1 (sFlt-1). Furthermore, excess sFlt-1 significantly increased oxidized-b2GPI secretion in HepG2 cells. Circulating oxidized-b2GPI levels were significantly higher in women with pre-eclampsia than in those without pre-eclampsia. Therefore, oxidized-b2GPI may contribute to the pathogenesis of pre-eclampsia. Under oxidative stress, the excessive oxidation of b2GPI and/or excessive placental sFlt-1 may trigger a negative spiral between trophoblast and liver cells.”

Line 111 – extravillous trophoblast cell line

We have added “extravillous” as below;

(Line 114 in the revised manuscript with changes)

“Human first trimester extravillous trophoblast, HTR-8/SVneo,…”

Line 158 and 159 – give the full names before introducing abbrevations TRX-1 and TRX-R, and briefly explain why they are added.

We have changed as below;

(Line 167 in the revised manuscript with changes)

…was preincubated with thioredoxin-1 (TRX-1) (3.5 uM) activated with thioredoxin reductase (TRX-R) (10 nM) plus nicotinamide adenine dinucleotide phosphate (NADPH; 200 µM) to generate free thiols within b2GPI.

Line 132 – The title is not correct, no cell proliferation method is described. It is also gramatically not correct.

We have deleted “in cell proliferation assays” and added the section “cell counting” as below;

(Line 135 in the revised manuscript with changes)

"Cell counting

Ten microliters of cell suspension were analyzed using a hemocytometer."

To what section do lines 389-394 belong? Should they be before Figure 5 lengend? This way this paragraph stands unlinked to other text.

(Line 407 in the revised manuscript with changes)

We have moved this part, it is now just before the figure legend for Fig 5.

Reviewer #2:

This article in the abstract and introduction sets the scene why this research was undertaken which is easy to follow. The researchers may also wish to briefly include a few lines either in their discussion or introduction the recent online publication in Nature 11 December 2024 by Kelsey L Swingle et al Placenta-tropic VEGF mRNA lipid nanoparticles ameliorate murine pre-eclampsia, which found that "an endogenous targeting mechanism based on β2-glycoprotein I adsorption that enables LNP delivery to the placenta." This seems to further support the hypothesis of this paper that beta 2 glycoprotein I plays an important biological role in placenta biology.

The methods and ethics and statistical methodology are sound

Results figures and tables are easy to follow and have a logical sequence

My only concern is with line 281-282 (an approximately 5 fold increase.. Fig 3A). When I look at Figure 3A the increase is much less than 5 fold, going from approximately 12.5 ng/ml of total B2GPI in control to 20ng/ml of the TRXI-TRXR treated and to 16ng/ml total B2GPI for the pCMV3-APOH without TRX1-TRXR treatment.

The discussion is easy to follow and ties the results together nicely.

It was an honour for us to have our manuscript reviewed by you, as we are admirers of your b2GPI work.

(Line 281-282 in original, Line 304 in the Revised Manuscript with Track Changes)

The error has been corrected as follows.

“The secretion of total-b2GPI significantly increased in the APOH-transferred groups (an approximately 5-fold increase, P = 0.006, P = 0.002, without or with TRX-1 treatment group, respectively. vs. control plasmid DNA-transferred group with no TRX treatment, P < 0.001, vs. control plasmid DNA-transferred group with TRX-1 treatment group; P < 0.001, Wilcoxon rank-sum test, Fig 3A).”

We mentioned the recent manuscript (Swingle KL et al., Nature 2025 Jan;637:412-421.), which you referred to, in the discussion part as follows;

(line 468 in the Revised Manuscript with Track Changes)

“Very recently, placenta-tropic lipid nanoparticles (LNPs) using a mechanism based on absorption with endogenous b2GPI have been reported [59]. The study demonstrated that the uptake of LNPs in the liver was significantly increased in b2GPI knockdown mice following intravenous siRNA administration, while it was reduced in the placenta [59]. Furthermore, a significant increase in blood b2GPI levels was observed in inflammation-induced PE model mice, and these mice showed a higher uptake of LNPs in the liver but not in the placenta. These results suggest the possibility of differential actions of b2GPI in the placenta and in the liver.”

---

## [Decision Letter · Decision Letter 1]

5 Mar 2025

Beyond antibodies: Beta-2 glycoprotein I as the unsung guardian of pregnancy

PONE-D-24-44937R1

Dear Dr. Nakamura,

We’re pleased to inform you that your manuscript has been judged scientifically suitable for publication and will be formally accepted for publication once it meets all outstanding technical requirements.

Kind regards,

María Teresa Llinás

Academic Editor

PLOS ONE

Additional Editor Comments (optional):

Reviewers' comments:

Reviewer's Responses to Questions

**Comments to the Author**

1. If the authors have adequately addressed your comments raised in a previous round of review and you feel that this manuscript is now acceptable for publication, you may indicate that here to bypass the “Comments to the Author” section, enter your conflict of interest statement in the “Confidential to Editor” section, and submit your "Accept" recommendation.

Reviewer #1: All comments have been addressed

Reviewer #3: All comments have been addressed

2. Is the manuscript technically sound, and do the data support the conclusions?

Reviewer #1: (No Response)

Reviewer #3: Yes

3. Has the statistical analysis been performed appropriately and rigorously? 

Reviewer #1: (No Response)

Reviewer #3: Yes

4. Have the authors made all data underlying the findings in their manuscript fully available?

Reviewer #1: (No Response)

Reviewer #3: Yes

5. Is the manuscript presented in an intelligible fashion and written in standard English?

Reviewer #1: (No Response)

Reviewer #3: Yes

6. Review Comments to the Author

Reviewer #1: (No Response)

Reviewer #3: (No Response)

7. PLOS authors have the option to publish the peer review history of their article (what does this mean? ). If published, this will include your full peer review and any attached files.

**Do you want your identity to be public for this peer review?** For information about this choice, including consent withdrawal, please see our Privacy Policy .

Reviewer #1: No

Reviewer #3: No

---

## [Editor Report · Acceptance letter]

PONE-D-24-44937R1

PLOS ONE

Dear Dr. Nakamura,

I'm pleased to inform you that your manuscript has been deemed suitable for publication in PLOS ONE. Congratulations! Your manuscript is now being handed over to our production team.

Kind regards,

on behalf of

Dr. María Teresa Llinás

Academic Editor

PLOS ONE